# Tracking the genome-wide occupancy of Arabidopsis LEAFY COTYLEDON1 in endosperm development
Jingpu Song [1,2,6], Xin Xie [3,4,6], Ioannis Mavraganis [1], Bianyun Yu [1], Wenyun Shen[1], Hui Yang[1], Daoquan Xiang [1], Yangdou Wei [2], Yuhai Cui [3,4] ✉ & Jitao Zou [1,5] ✉

Endosperm development is crucial for embryo growth and seed maturation. *LEAFY COTYLEDON1* (*LEC1*), expressed in both endosperm and embryo, serves as a key regulator of seed development, orchestrating processes such as embryogenesis and seed maturation. *LEC1* expression in the endosperm is detectable within a day after fertilization, yet its specific regulatory networks and developmental functions in this tissue remain unclear. To address this, we employed a modified INTACT system to isolate endosperm nuclei and performed ChIP-seq to map the genome-wide binding profile of LEC1 in developing endosperm. Integrating ChIP-seq with transcriptomic analyses, we uncover a critical role for LEC1 in regulating diverse biological pathways. Differential gene expression analysis in the endosperms of *lec1* mutant and wild type shows substantial changes, particularly in genes involved in secondary cell wall biogenesis, photosynthesis, and lipid metabolism. Notably, LEC1's regulatory networks in the endosperm shift significantly after cellularization, with distinct genes being activated in the cellular and degeneration stages. The absence of *LEC1* causes significant alterations in endosperm metabolism, particularly affecting storage lipid fatty acid composition. These findings provide insights into the essential role of *LEC1* in endosperm development and its broader impact on seed formation.

The unique double fertilization event in angiosperms gives rise to both the embryo and the endosperm within a seed[1]. The endosperm is mainly employed by flowering plants as a supporting tissue to sustain embryo development, gradually consumed by the growing embryo during seed development[1,2]. In some plant species, such as cereals, endosperm serves as a crucial carbon and energy reservoir and remain persistent until seed germination[2]. Endosperm development is an autonomously programmed process independent of embryogenesis; however, establishing a viable seed requires a rigorously temporal execution of endosperm development[3–5]. In *Arabidopsis thaliana* (Arabidopsis), after fertilization the central cell (primary endosperm) rapidly undergoes nuclear proliferation without cytokinesis[6]. A recent report reveals that CYCD7;1, a cell division factor, functions as a paternal signal to inform the central cell of a fertilization occurrence, thereby initiating nuclei proliferation (NP)[7]. The phytohormone auxin triggers endosperm nuclear replication but prevents cellularization[5]. Critical genes regulating auxin biosynthesis in the central

cell, most notably *YUCCA10* (*YUC10*), are repressed by FERTILIZATION INDEPENDENT SEED-Polycomb Repressive Complex 2 (FIS-PRC2) prior to fertilization and the repression is lifted in primary endosperm through switching off the expression of *ATHB* genes by AGAMOUS-LIKE 62 (AGL62)[5,8]. Approximately 4 days after pollination (DAP), the *AGL62*-promoted auxin biosynthesis is then repressed by FIS-PRC2 to initiate cellularization[4,6,9]. The timing of the transition from nuclear proliferation to cellularization determines the final seed size[10]. Endosperm cellularization establishes dehydration tolerance in the developing embryo through ABA biosynthesis and signaling to ensure embryo survival during seed maturation[11]. After endosperm cellularization completes at around 6 DAP, embryo grows rapidly to the space made available by the diminishing endosperm and accumulates seed storage products[12]. Embryo expansion leads to developmentally controlled programmed cell death (degeneration) of endosperm, governed by multiple NAC transcription factors in the embryo-adjacent endosperm[13,14]. ZHOUPI-dependent endosperm

[1]Aquatic and Crop Resource Development Research Centre, National Research Council of Canada, Saskatoon, SK, Canada. [2]Department of Biology, University of Saskatchewan, Saskatoon, SK, Canada. [3]London Research and Development Centre, Agriculture and Agri-Food Canada, London, ON, Canada. [4]Department of Biology, Western University, ON, CanadaLondon. [5]Department of Plant Science, The Pennsylvania State University, University Park, PA, USA. [6]These authors contributed equally: Jingpu Song, Xin Xie. ✉e-mail: yuhai.cui@agr.gc.ca; jpz5606@psu.edu

weakening also facilitates the embryo invasion process[15,16]. At maturation, only the peripheral endosperm layer remains intact and alive, playing a key role in coordinating and promoting seed germination[13].

LEAFY COTYLEDON1 (LEC1) is a central regulator in seed development[17–19]. Mutations in *LEC1* cause defective seed phenotypes, such as arrested embryo development with short embryo axis, less developed cotyledons with anthocyanin accumulation, and desiccation intolerance[18,20]. LEC1 acts in combination with other transcription factors (i.e., AREB3, bZIP67, and ABI3) to regulate distinct gene sets in diverse embryo developmental processes[17]. A transcriptome profiling study of *lec1* mutant Arabidopsis seeds highlights multiple gene networks and developmental processes controlled by LEC1 in spatially and temporally controlled patterns[18]. *LEC1* is expressed in the endosperm of many species, including *Brassica napus*, Arabidopsis, and rice[20–23]. Endosperm-originated LEC1 is sufficient for embryo maturation in the absence of embryo-sourced *LEC1* expression[20]. However, aside from its function of supporting embryo development and maturation, whether the endosperm-synthesized LEC1 has any roles in the development and functionality of the endosperm remains unanswered.

To extend our understanding of the role of *LEC1* in the endosperm, we have implemented a modified isolation of nuclei tagged in specific cell types (INTACT) system[24,25], in combination with the chromatin immunoprecipitation (ChIP) followed by next generation sequencing (ChIP-Seq) assays, to investigate LEC1 genome-wide occupancy in endosperm. Transcriptome analyses of *lec1* endosperm indicates that LEC1 is required to modulate expression of genes regulating key biological processes in the endosperm development, including auxin biosynthesis and signaling, sugar transport, ABA signaling, and cell wall modification. Moreover, lack of *LEC1* alters endosperm storage lipid biosynthesis. Our findings shed light on the regulatory role of LEC1 in endosperm development.

## Results

### Genome-wide identification of LEC1-occupied genes in endosperm

In a previous study, we generated a *lec1* complementary line, *lec1-1 pLEC1:LEC1-GFP* (PLL), which restores *LEC1* expression levels (Supplementary Fig. 1) and successfully rescues the seed defects observed in the *lec1-1* mutant[20]. To determine whether LEC1 protein is present in the endosperm, we introduced an endosperm nucleus reporter *pPHE1::mCherry-NLS* into the PLL background, generating a double reporter line. Notably, *PHE1* is an endosperm-specific gene, but its promoter activity declines sharply upon cellularization[25]. We examined the developing seeds at 3 DAP and observed that LEC1-GFP signals co-localized with the endosperm nucleus marker mCherry-NLS, confirming the presence of LEC1 in endosperm nuclei (Fig. 1A).

With LEC1's presence in the endosperm confirmed, we sought to identify its potential targets and biological functions specifically in this tissue. A critical step for ChIP-seq assays was obtaining a sufficient quantity of endosperm nuclei. However, due to the technical challenges of isolating large amounts of endosperm nuclei without tissue contamination from manual dissection, we modified the original INTACT construct[24] and developed two constructs, *PIM* (*pACT2::TurboID pPHE1::mCherry-WPP-BLRP*) (Fig. 1B) and *ZIM* (*pACT2::TurboID pZOU::mCherry-WPP-BLRP*) (Supplementary Fig. 2A). Unlike *PHE1*, which is highly expressed in early endosperm development, *ZHOUPI* (*ZOU*) is predominantly active during endosperm CE and degeneration (DE)[13,15,16]. We then introduced *PIM* and *ZIM* into the PLL background, generating PLL-PIM and PLL-ZIM lines, respectively. Immunoblotting and nuclei-beads binding assays confirmed the technical feasibility of these constructs for endosperm nuclei isolation (Fig. 1C, D, Supplementary Fig. 2B, C). Following an established INTACT-based protocol[25], we isolated endosperm nuclei from developing seeds for ChIP-Seq assays on LEC1-GFP (Fig. 1E). Specifically, we collected seeds at 1–3 DAP from PLL-PIM plants to obtain nuclei at the nuclei proliferation (NP) stage, whereas we harvested seeds at 5 and 7 DAP from PLL-ZIM plants to isolate nuclei at the cellularization (CE), and degeneration (DE) stages, respecitvely. For each stage, ChIP-Seq experiments were conducted with two biological replicates, which showed strong correlations (Supplementary Fig. 3A, B).

Data analyses revealed 8517 LEC1-occupied genomic sites (LOGS) at the NP stage (Supplementary Table 1), primarily distributed along the chromosome arms and enriched in promoter regions (Fig. 1F, Supplementary Fig. 3C). These LOGS corresponded to 7379 genes (Supplementary Data 1), with significant enrichment near transcription start sites (Fig. 1G, Supplementary Fig. 3D), suggesting a key role for LEC1 in transcriptional regulation within the endosperm. Among the identified LOGS-associated genes, several well-characterized endosperm development regulators, *FIE*, *VIM1*, *VIM2 YUC10*, and *AGL61*, were selected for ChIP-Seq data validation using independent ChIP-qPCR assays (Fig. 1H, I). *bZIP62*, which was not identified in the LOGS dataset, was used as a non-LEC1 binding control for validation (Fig. 1H, I). Comparatively, 1721 and 1724 LOGS were detected at the CE and DE stages, respectively (Supplementary Table 1). The unexpectedly low number of LOGS at CE and DE led us to focus our research on the NP-stage ChIP-seq dataset. Collectively, our ChIP-seq data provide a comprehensive list of potential LEC1 targets, enabling the indentification of direct LEC1-regulated genes in the endosperm.

### LEC1 transcriptionally regulates auxin biosynthesis, transport, and signaling in syncytial endosperm

We perfomed RNA-Seq analyses to explore the transcriptome profiles of developing endosperms in wild type (WT) and *lec1-1* mutant at the NP, CE, and DE stages (Supplementary Table 2). Principal component analysis revealed that samples from the same developing stage clustered together, with the NP transcriptome distinct from those of CE and DE stages (Supplementary Fig. 4A). Notably, we retrieved embryo-specific and seed coat-specific genes from previous studies[22] and found that their average transcript levels were not significantly enriched in any of the datasets (Supplementary Fig. 4B–D). This suggests minimal tissue contamination in the endosperm RNA samples. Differentially expressed genes (DEGs) were defined as those with at least 1.5-fold change in expression (up or down) between *lec1* mutant and WT endosperms, with statistical significance ($p < 0.05$). At NP, 887 down-regulated and 2501 up-regulated genes were identified in *lec1-1* endosperms (Fig. 2A, Supplementary Data 2). Of these, 335 down-regulated and 641 up-regulated genes overlapped with those identified as LEC1-occupied genes at NP in the aforementioned ChIP-Seq analysis (Fig. 2B, Supplementary Data 3). These LEC1-occupied DEGs were classified as LEC1 target genes (LTGs).

Gene ontology (GO) analysis showed that the 335 down-regulated LTGs were overrepresented in the positive regulation of transcription by RNA polymerase II (Fig. 2C). This group included 12 *AGLs* (Supplementary Fig. 5A), many of which are endosperm-specific, such as *AGL36*, *AGL46*, *AGL62*, *AGL90*, and *AGL102*[26,27] (Fig. 2D). AGL62 plays a critical role in endosperm proliferation and the transition to CE through regulating genes involved in auxin biosynthesis, particularly *TAA1*, *TAR1*, and *YUC10*[5,8,9]. These genes were down-regulated in the *lec1* endosperm (Supplementary Fig. 5B), suggesting LEC1's direct involvement in modulating the AGL62-YUC10 regulatory network for auxin biosynthesis. Furthermore, *YUC11*, another endosperm-specific YUC gene, was also down-regulated in the *LEC1* endosperm, whereas *YUC1* and *YUC4*, which are not specific to the endosperm specific genes, were not detected in either *lec1-1* or WT endosperms (Supplementary Fig. 5B). These results support the hypothesis that LEC1 directly regulates a subset of endosperm-expressed *AGLs* to control endosperm-specific developmental processes. We also investigated the biological processes of non-LTGs at the NP stage. GO analysis showed enrichment in photosynthesis, gluconeogenesis, and fructose metabolic process (Supplementary Fig. 5C). Previous reports have indicated LEC1's role in regulating genes related to photosynthesis and chloroplast biogenesis in Arabidopsis seeds[18]. Consistent with this, we found a significant reduction in genes associated with photosynthesis in *lec1* mutant endosperms (Supplementary Fig. 5D), highlighting LEC1's involvement in conserved biological processes between the endosperm and embryo.

The 641 up-regulated LTGs were significantly enriched for GO terms related to sucrose transport, response to auxin, regulation of cell wall, regulation of cell differentiation, and positive regulation of auxin biosynthesis

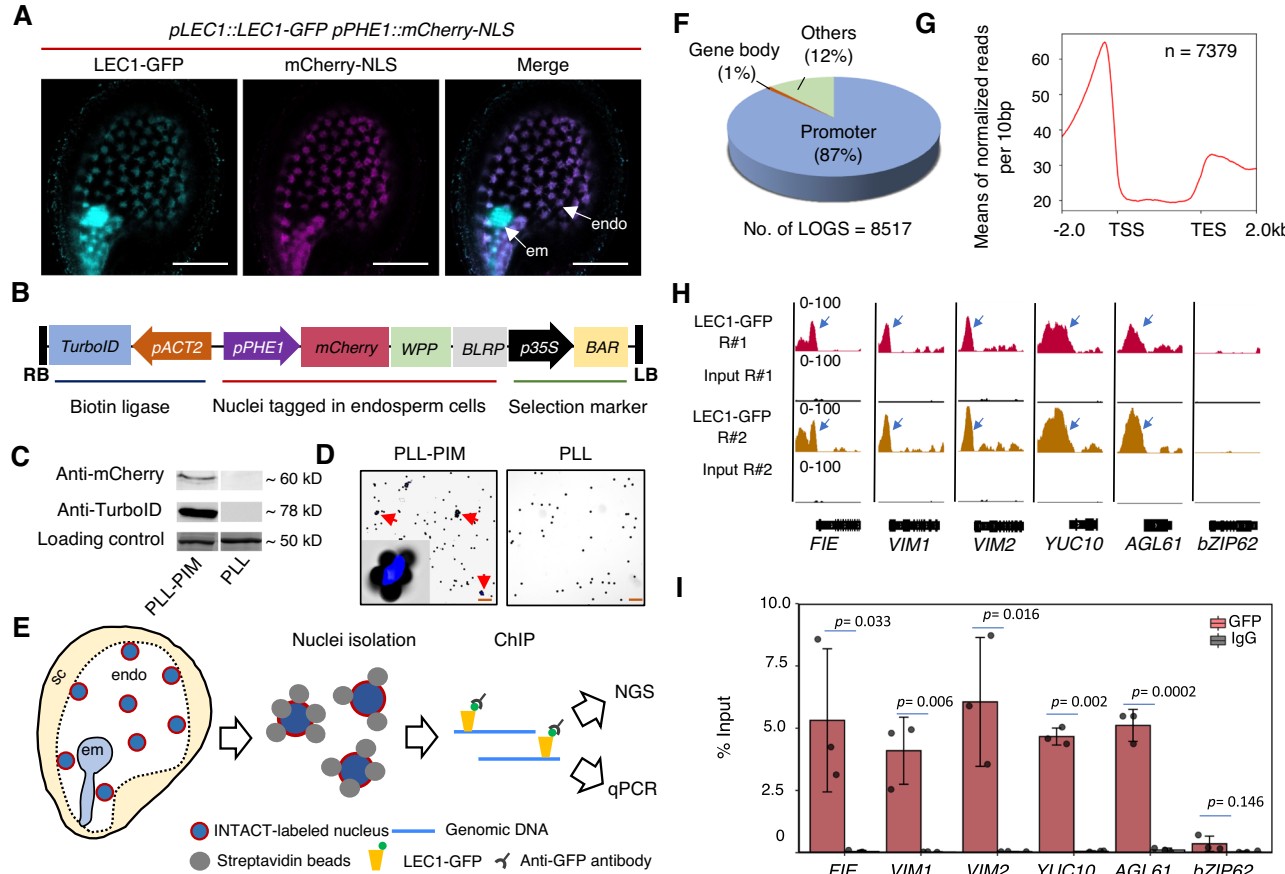

**Fig. 1 | Genome-wide profiling of LEC1 occupancy in developing endosperm.**
**A** Co-localization of LEC1-GFP and endosperm-expressed mCherry-NLS in the developing seed at 3 day after pollination (DAP). Asterisks indicate nuclei. GFP signals are shown in cyan, mCherry signals are shown in magenta. em embryo, endo endosperm. Scale bar: 100 μm. **B** Schematic diagram showing the transgene structure of modified INTACT construct PIM used for plant transformation. **C** Immunoblot showing the signals of mCherry (two bands, upper: biotinylated mCherry-WPP-BLRP lower band: non-biotinylated protein) and TurboID proteins in the developing seeds of the transgenic line PLL-PIM. PLL seeds were used as negative control. The images are representative of three independent replicates. **D** Binding assay of beads-bound nuclei and free beads from PLL-PIM and PLL developing seeds. Red arrows indicate beads-bound nuclei. Insets: magnified individual nuclei binding beads. Scale bar: 200 μm. **E** Schematic workflow showing nuclei purification from NP (nuclei proliferation) endosperm tissues targeting for chromatin immunoprecipitation (ChIP) followed by Next Generation Sequencing (NGS) and qPCR validation. sc seed coat, em embryo, endo endosperm. **F** Pie graph illustrating the percentage distribution of LOGS (LEC1-occupied genomic sites)

across annotated genic and intergenic regions in endosperms. **G** Mean density of all LOGS-corresponding genes in endosperm. Plotting regions were scaled to the same length as follow: 5' ends (−2 kb to transcription start site (TSS) and 3' ends to transcription termination site (TTS) to downstream 2 kb) were not scaled, and the gene body was scaled to 3 kb. **H** IGV views of two independent replicates of ChIP-Seq signals on five representative genes from LEC1-occupied genes at NP. *bZIP62* is shown as a non-LEC1-occupancy gene. LEC1-occupancy peaks at gene promoter regions are indicated by arrow heads. Gene structures are shown underneath each panel. The orientation is from START to END positions as left to right. The y-axis scales are the number of normalized ChIP-Seq counts for every 10 bp window. LEC1-GFP tracks (two biological replicates) are shown in red and brown. Input track is shown in black. **I** ChIP-qPCR validation of LEC1 occupancy at targets shown in (**H**). Data are shown as the percentage of input. Anti-IgG was used as the negative control samples, and the *bZIP62* locus was used as the negative control locus. Error bars are presented as mean values ±S.E. from three biological replicates (*n* = 3). *p* values were determined by conducting Student's *t* test.

(Fig. 2E). Among these, we found several LTGs related to auxin transport and signaling. As shown in Fig. 2F and Supplementary Fig. 6A, genes encoding the transcription factor IDD14[28], as well as three auxin transporters WAT1[29], AUX1[30], PIN1[31], and downstream auxin-signaling proteins from the IAA and GH3 families[32], were significantly up-regulated in *lec1-1* endosperms. Additionally, another set of auxin transporter and auxin-responsive genes were indirectly up-regulated by LEC1 (Supplementary Fig. 6B, C). At the NP stage, sugars are taken up by the endosperm from the integuments and subsequently re-exported to the embryo surface through transmembrane sucrose transporters[14]. We observed a significant up-regulation of three *SWEET* sugar transporters (*SWEET10*, *SWEET11*, and *SWEET12*) and two SUC sucrose exporters (*SUC1* and *SUC2*) in the *lec1* endosperms compared to WT (Fig. 2G), suggesting that LEC1 is involved in regulating sugar flow from the endosperm to the embryo. One hallmark of syncytial endosperm is the absence of cytokinesis during nuclear divisions[1]. Interestingly, lack of *LEC1* in the syncytial endosperm led to up-regulation of several secondary cell wall

biosynthesis master regulators, such as *MYB20*, *MYB43*, *NAC026*, *NAC075*, and *NAC105* (Fig. 2G)[33], implying that LEC1 plays a role in repressing cell wall formation during endosperm proliferation phase.

As shown in Fig. 2H, LEC1 directly and/or indirectly activates well characterized endosperm-expressed auxin biosynthesis-related genes (i.e., *AGL62* and *YUC10*), while paradoxically down-regulating genes involved in auxin transport and signaling during the endosperm nuclear proliferation stage. Moreover, LEC1 regulates genes involved in photosynthesis, sugar transport, and cell wall formation.

### Evidence for a major shift of LEC1-driven regulatory networks in the endosperm after CE
We performed the DEG analyses for the RNA-seq datasets of the CE and DE stages to explore whether distinct target genes are activated at different stages of endosperm development. Transcriptomic analysis revealed 1223 down-regulated and 1302 up-regulated DEGs at the CE stage (Fig. 3A,

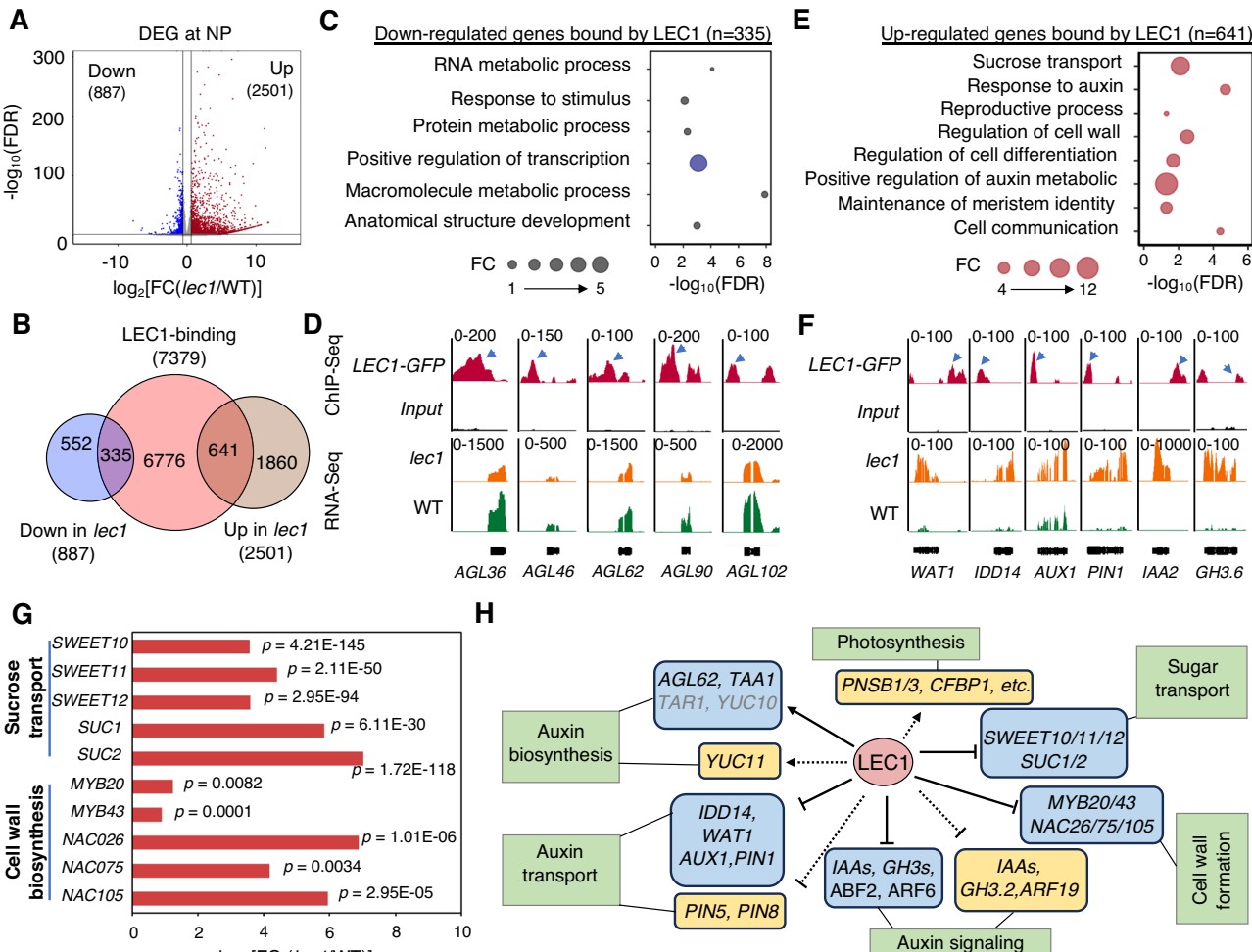

**Fig. 2 | LEC1 directly regulates endosperm auxin-related gene expression at nuclei proliferation stage. A** Volcano plot showing down-regulated and up-regulated genes in *lec1* endosperm compared to that of WT (Fold change, FC ≥ 1.5, FDR < 0.05). The *x*-axis is presented as $\log_2$(FC) value by calculating fragments per kilobase of transcript per million mapped reads (FPKM) in *lec1* versus the WT; the y axis represents the $-\log_{10}$ of the FDR (false discovery rate). **B** Venn diagram indicates the overlaps between genes occupied by LEC1 (ChIP-Seq) and genes with differential expressions (RNA-Seq) at NP. **C** GO analysis results elucidating the over-represented biological process (highlighted in navy blue color) enriched among the LEC1-occupied down-regulated genes in *lec1* endosperm at NP. **D** Integrative Genomics Viewer (IGV) views of ChIP-Seq signals and RNA-Seq signals on representative genes from LEC1-occupied down-regulated genes. LEC1-occupancy

peaks at gene promoter regions are indicated by arrow heads. The y-axis scales are the number of normalized ChIP-Seq or RNA-seq counts for every 10 bp window. **E** GO analysis results elucidating the over-represented biological process enriched among the LEC1-occupied up-regulated genes in *lec1* endosperm at NP. **F** IGV views of ChIP-Seq signals and RNA-Seq signals on representative genes from LEC1-occupied up-regulated genes. LEC1-occupancy peaks at gene promoter regions are indicated by arrow heads. The y-axis scales are the number of normalized ChIP-Seq or RNA-seq counts for every 10 bp window. **G** Gene expression levels of LEC1-occupied up-regulated genes involved in sucrose transport and cell wall biosynthesis. **H** Schematic of LEC1 direct and indirect targets and downstream loci with documented roles in early endosperm development.

Supplementary Data 2). We next investigated whether these DEGs observed in the cellular endosperms were directly regulated by LEC1. A Venn diagram showed that 423 down-regulated and 314 up-regulated LTGs were identified at CE (Fig. 4B, Supplementary Data 3). The 423 down-regulated LTGs were overrepresented for positive regulation of small molecule metabolic process, including genes, such as *EFR055*, *FUS3*, *NFYB9*, *MYB115*, *MYB118*, and *WAT1* (Fig. 4C, D). The 314 up-regulated LTGs were predominantly involved in gibberellin catabolic and metabolic processes, such as *bHLH93* and *GA2OX2/3/4/5*, along with genes implicated in other processes, including xenobiotic export from cell, shoot and organ development, regulation of transcription, glutamine family metabolic, brassinosteroid metabolism, and homeostasis (Fig. 4E, F).

In comparison to the CE stage, we identified slightly more DEGs at the DE stage, with 1465 down-regulated and 1397 up-regulated genes (Fig. 3G, Supplementary Data 2). Similarly, we identified 390 down-regulated and 429 up-regulated LTGs at DE (Fig. 3H, Supplementary Data 3). The

390 down-regulated LTGs were mainly enriched in biological processes related to plant organ and seed development, somatic embryogenesis, and post-embronic morphogenesis, respond to lipid, hormone signaling, and abaxial/adaxial cell specification (Fig. 3I). Among these overrepresented GO terms, one group of genes including *VAL1*, *GATA21*, *MYB118*, and *PP2C54*, were associated with seed development regulation, while another group, including *YAB1*, *YAB2*, *KAN2*, and *JAG*, were involved in abaxial cell fate specification (Fig. 3J). The 429 up-regulated LTGs were significantly enriched for the GO term glucosinolate catabolic process, including genes such as *BGLU20*, *BGLU21*, *BGLU22*, *BGLU23*, and *ESP*, as well as other processes, including xenobiotic export from cell, plant organ development, respond to lipid, and regulation of transcription (Fig. 3K, L).

Together, these findings show that LEC1 regulates a set of biological processes in cellular endosperms that are different from those in syncytial endosperm, highlightling a major shift in LEC1-driven regulatory networks following endosperm cellularization.

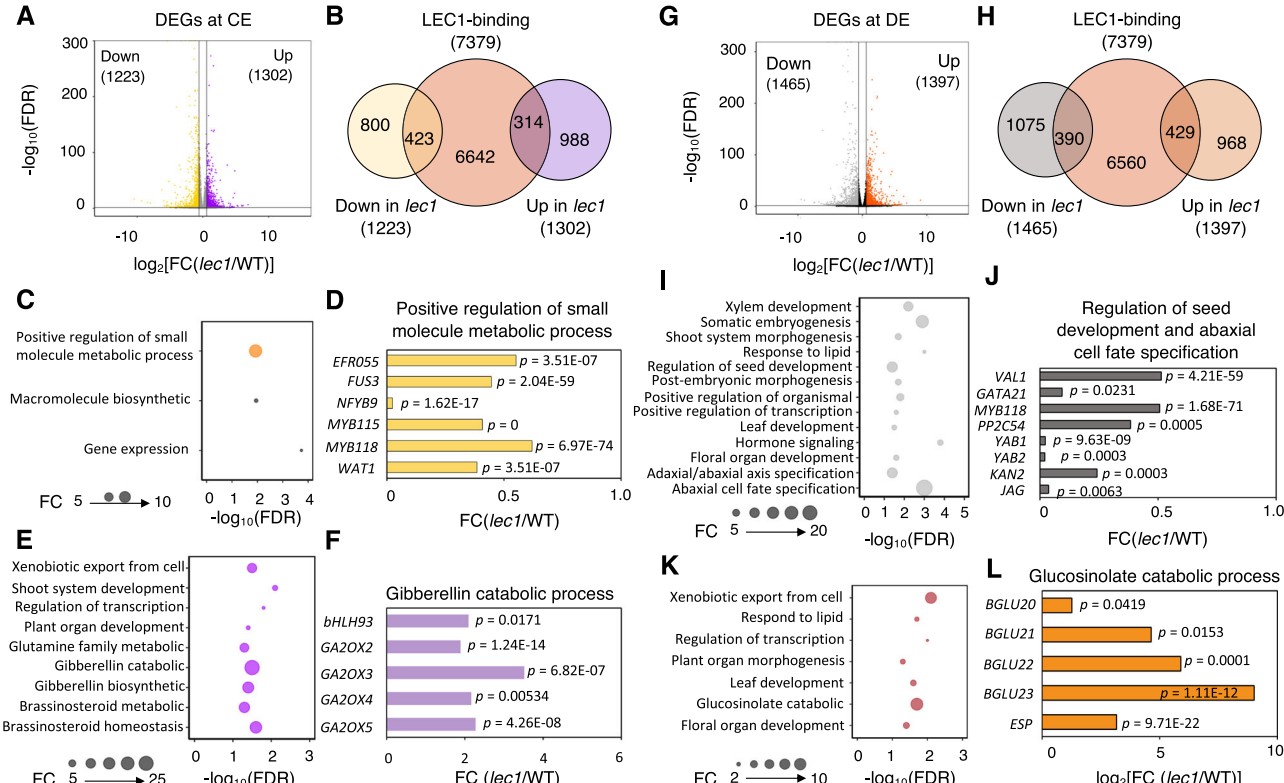

**Fig. 3 | Identification of LEC1 targets after endosperm cellularization. A** Volcano plot showing down-regulated and up-regulated genes in *lec1* endosperm at CE compared to that of WT (FC ≥ 1.5, FDR < 0.05). The *x*-axis is presented as $\log_2$(FC) value by calculating FPKM in *LEC1* versus the WT; the y axis represents the $-\log_{10}$ of the FDR. **B** Venn diagram shows the overlaps between genes occupied by LEC1 (ChIP-Seq) and genes with differential expressions (RNA-Seq) at CE. **C** GO analysis results elucidating the over-represented biological process (highlighted in bright yellow color) enriched among the down-regulated genes by LEC1 in *lec1* endosperm at CE. **D** Gene expression levels of down-regulated genes involved in biological process "positive regulation of small molecule metabolic process" at CE. The x-axis value is shown as fold change of *lec1* versus WT. **E** GO analysis results elucidating the over-represented biological process (highlighted in purple color) enriched among the up-regulated genes by LEC1 in *lec1* endosperm at CE. **F** Gene expression levels of the up-regulated genes involved in biological process "gibberellin catabolic processes" at CE. The *x*-axis value is shown as FC of *lec1* versus WT. **G** Volcano plot

showing down-regulated and up-regulated genes in *lec1* endosperm at DE compared to that of WT (FC ≥ 1.5, FDR < 0.05). The x-axis is presented as $\log_2$(FC) value by calculating FPKM in *lec1* versus the WT; the y axis represents the $-\log_{10}$ of the FDR. **H** Venn diagram indicates the overlaps between genes occupied by LEC1 (ChIP-Seq) and genes with differential expressions (RNA-Seq) at DE. **I** GO analysis results elucidating the over-represented biological process (highlighted in gray color) enriched among the down-regulated genes by LEC1 in *lec1* endosperm at DE. **J** Gene expression levels of LEC1-occupied down-regulated genes involved in biological process "regulation of seed development and abaxial cell fate specification" at DE. The *x*-axis value is shown as FC of *lec1* versus WT. **K** GO analysis results elucidating the over-represented biological process (highlighted in brown color) enriched among the LEC1-occupied up-regulated genes in *lec1* endosperm at DE. **L** Gene expression levels of NP-LEC1-occupied up-regulated genes involved in biological process "glucosinolate catabolic process" at DE. The *x*-axis value is shown as $\log_2$(FC) of *lec1* versus WT.

## Analyses of LTGs in developing embryo and endosperm

A previous study identified 554 LTGs in the Arabidopsis embryo (Supplementary Data 4)[18]. To investigate whether LTGs are conserved between the endosperm and embryo, we compared the LTGs in both tissues. The Venn diagram showed that the vast majority (87%) of embryo-LTGs were not found in the endosperm at any stage, whereas 574 out of 976, 343 out of 737, and 387 out of 819 LTGs were unique to the NP, CE, and DE stages of the endosperm, respectively (Fig. 4A, Supplementary Data 4). Additionally, 74 LTGs were shared between the embryo and endosperm (Supplementary Data 4). The large difference in LTGs between the embryo and endosperm suggests that LEC1 plays distinct roles in seed development at different spatial locations. To further verify this conclusion and explore LEC1's diverse functions in seed development, we performed GO analyses on the LTGs that were embryo-specific, endosperm (NP, CE, and DE)-specific, and common to both embryo and endosperm. The overrepensed GO terms for each group are displayed in Fig. 4B. As previously reported, LTGs in embryo were mostly enriched for GO terms related to cell wall modification, photosynthesis, and respond to abiotic stimulus[18]. In the endosperm, NP-specific LTGs were primarily enriched for GO terms related to secondary cell wall biogenesis, glucosinolate metabolic, regulation of growth, and respond to lipid; however, CE- and DE-specific LTGs were enriched for GO terms

related to ABA metabolism and tissue development. The 74 LTGs shared between the embryo and endosperm were involved in fatty acid biosynthesis, somatic embryogenesis, and seed dormancy. While LEC1 is involved in conserved biological processes in both the embryo and endosperm, the roles it plays in each tissue and at different stages are quite distinct.

In soybean seeds, LEC1 regulates different gene sets through combinatorial interactions with other transcription factors, mediated by *cis*-regulatory modules containing clustered *cis* elements[17]. To further understand the mechnisms underlying LEC1's regulatory network in different tissues, we sought to identify overrepresented DNA motifs in the bound regions of LTGs. Figure 4C displays the DNA sequence motifs that were enriched in endosperm-LTGs from this study and those identified in embryo from the previous study[18]. The MADS, BPC1, REM19/VRN1, and ABI3 *cis*-regulatory elements were significantly overrepresented in all endosperm LTGs across the different stages. The BPC1 motif was the only one also identified in embryo-LTGs. Notably, the CCTTA-binding sequence and *G*-box motifs, which were reported in the embryo, were not significantly enriched in endosperm. The differential enrichment of DNA sequence motifs in LTGs between tissues further supports the idea that LEC1 regulates distinct sets of genes in different tissues, likely through combinatorial interactions with other transcription factors.

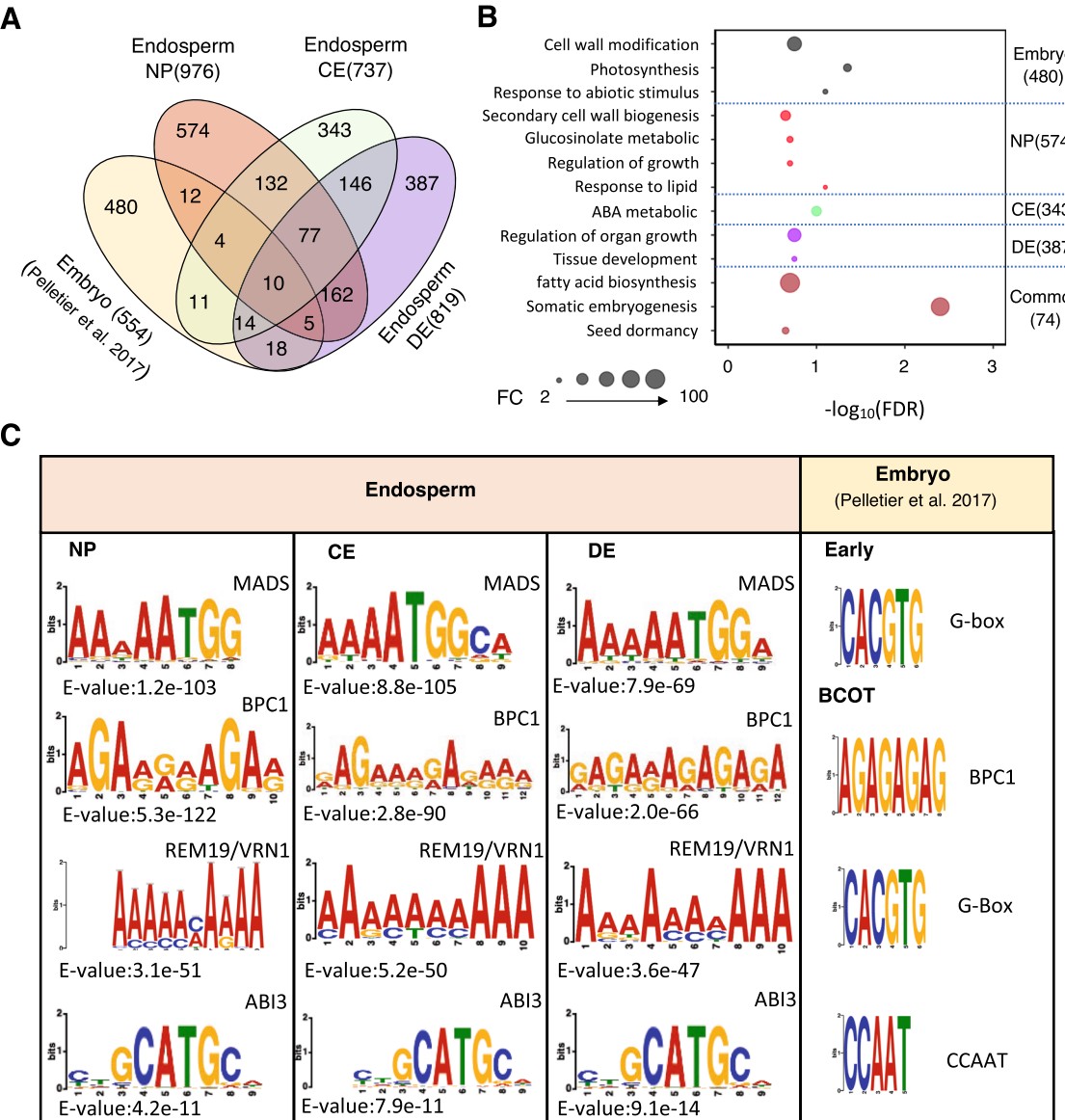

**Fig. 4 | Comparison of LEC1 targets between endosperm and embryo. A** Venn diagram shows the overlaps of LEC1 targets in embryo and endosperm. Gene list of LEC1 targets in embryo was retrieved from previous study. **B** GO analysis results elucidating the over-represented biological processes enriched among the LEC1 targets in embryo and endosperm (NP, CE, and DE). **C** The *cis*-elements over-represented (*E*-value < 0.05) in LEC1 targets identified in endosperm (NP, CE, and DE) and in embryo (collected from previous study).

## *lec1* mutant endosperm displays deficiencies in cell elimination and regulation of storage lipid biosynthesis

Loss-of-*LEC1* results in embryo arrest during the developmental transition from endosperm CE to endosperm DE, yet no clear morphological defects in the *lec1* endosperm have been described previously[18,20]. In Arabidopsis WT seeds, endosperm elimination outpaces endosperm proliferation at 8 DAP and eventually only a single layer of peripheral endosperm remains at nine DAP[13]. In contrast, the majority of *lec1-1* mutant endosperm persisted even at ten DAP (Fig. 5A). Notably, the morphological defects in the *lec1-1* endosperm were restored in the PLL seeds (Supplementary Fig. 7). The results indicate a delay in the endosperm elimination process in *lec1* seeds.

The single layer of endosperm stores carbon in the form of tria-cylglycerol (TAG), which constitutes approximately 10% of the total seed oil in mature seeds[34]. LEC1 is an important component of the regulatory network controlling oil content and composition in seed[18]. We dissected the embryo and the endosperm/seed coat fractions from the WT, *lec1-1*, and PLL seeds (Fig. 5B) and assessed their TAG contents. As expected, oil yield from *lec1-1* embryo was severely compromised, while PLL embryos

produced TAG comparable to the WT control (Fig. 5C). TAG level in the *lec1-1* endosperm was appreciably higher than in both the WT and PLL endosperms (Fig. 5D). Multiple compounding components likely contribute to this phenotype, but we hypothesize that the reduced resource demand from the embryo and increased resource allocation to the endosperm in *lec1* seeds—due to defects in prolonged endosperm elimination and embryo maturation—are key contributors.

The fatty acid profiles of the endosperm TAG are characteristically different from the embryos, and highly enriched with cis-ω-7 fatty acids, including the cis-ω-7 C16:1 (palmitoleic acid), the cis-ω-7 C18:1 (vaccenic acid), and the very long chain fatty acid (VLCFA) cis-ω-7 C20:1 (paullinic acid)[34]. Next, we analyzed the fatty acid compositions of TAG in the endosperm of WT, *lec1-1*, and PLL (Supplementary Fig. 8). There was no significant difference in total cis-ω-7 fatty acid content in the *lec1-1* endosperm when it is compared with the WT and the PLL (Fig. 5E). However, we detected drastic changes in the proportion of the cis-ω-7 C20:1, leading to a lowered ratio of cis-ω-7 C20:1 versus cis-ω-7 C18:1 in the TAG of the *lec1-1* endosperm (Fig. 5F). To confirm that *LEC1* is responsible for defects in seed

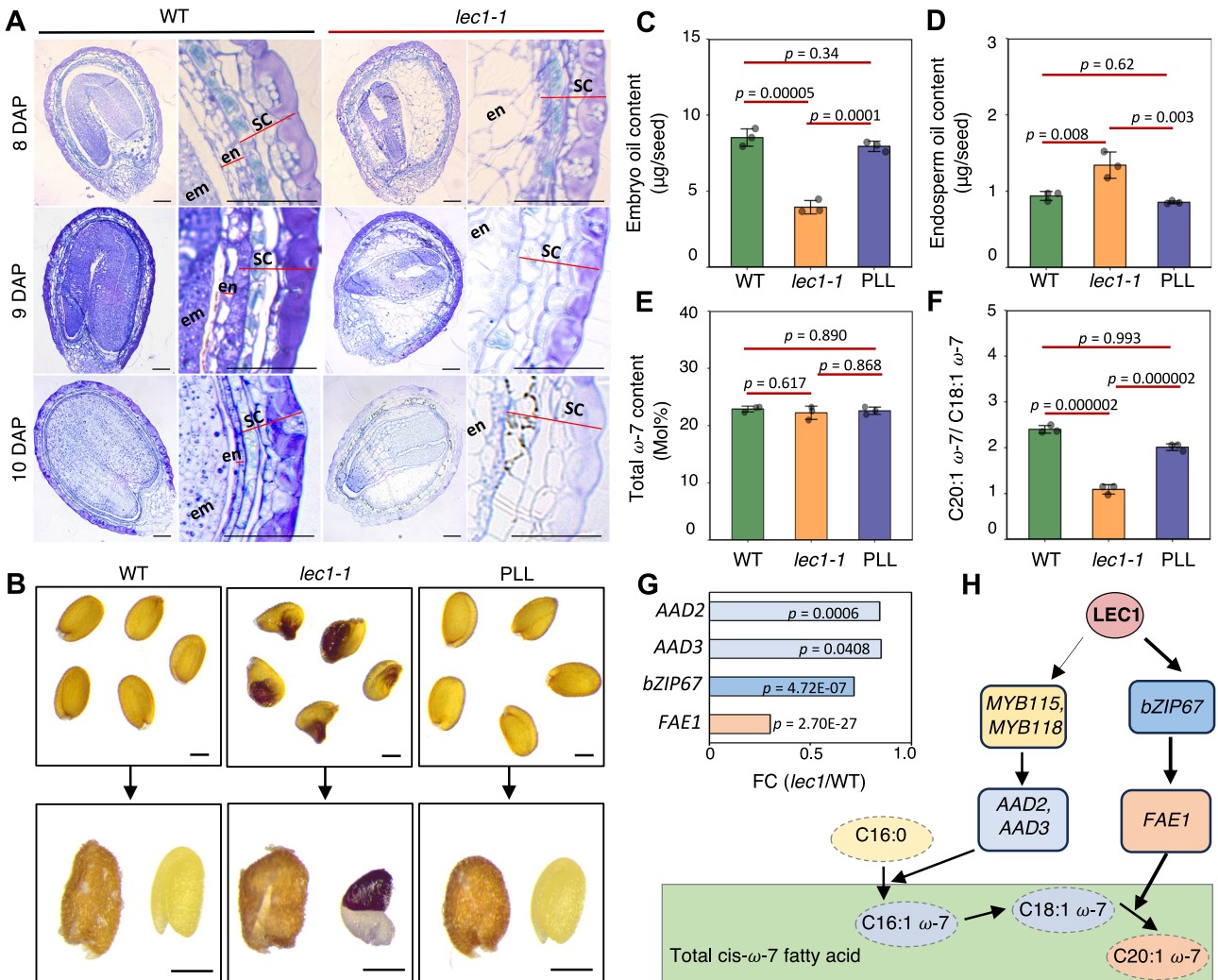

**Fig. 5 | Lack of LEC1 alters endosperm fatty acid composition. A** Semi-thin sections of 8–10 DAP seeds from WT and *lec1-1*. SC, seed coat; en, endosperm; em, embryo; scale bars: 50 μm. **B** Images are representative of WT, *lec1-1*, and PLL seeds used for tissue dissection and fatty acid content measurements. Scale bars: 200 μm. Bar graphs showing the total oil content per seed in the embryo (**C**) and in the endosperm (**D**). **E** Total cis-ω-7 content accumulated in the endosperms of WT, *lec1-1*, and PLL seeds. **F** Statistic analysis of the ratios of C20:1 ω-7 versus C18:1 ω-7 in the endosperms of WT, *lec1-1*, and PLL seeds. **C–F** Three independent replicates

were conducted to calculate the means of values in each figure. Error bars are presented as mean values ±S.E. from three biological replicates (*n* = 3). For each replicate, 15 seeds of each genotype were dissected to separate embryo and endosperm tissues. *p*-values were determined with one-way ANOVA followed by the post-hoc Tukey multiple comparison tests **G** Expression levels of genes in regulating cis-ω-7 fatty acid accumulation and elongation in *lec1-1* endosperm at DE. The x-axis value is shown as log$_2$ (FC) of *lec1* versus WT. **H** Schematic of LEC1 regulates cis-ω-7 fatty acid accumulation and elongation.

oil content, we analyzed seeds from the *lec1-3*, a T-DNA mutant of *LEC1* in the Landsberg (Ler) background, along with WT Ler and the *pLEC1::LEC1-GFP lec1-3* (LLL) plants[20]. Similar to *lec1-1*, *lec1-3* exhibited elevated endosperm oil content and a significantly reduced ratio of cis-ω-7 C20:1 to cis-ω-7 C18:1, both of which were restored in the complementary line LLL (Supplementary Fig. 9A–D). In Arabidopsis, two endosperm-expressed Delta-9 palmitoyl-ACP desaturases *AAD2* and *AAD3*, controlled by two MYB transcription factors MYB115 and MYB118, are responsible for generating the cis-ω-7 double bond[35]. The cis-ω-7 C18:1-CoAs are elongated to C20:1-CoA by FATTY ACID ELONGASE 1 (FAE1) directly regulated by bZIP67[36,37]. Lack of *FAE1* failed to produce cis-ω-7 C20:1 in Arabidopsis seed endosperm (Supplementary Fig. 10). *MYB115*, *MYB118*, and *bZIP67* were identified as LTGs whereas no binding signal of LEC1 was enriched in the genomic regions of either *AAD2* or *AAD3* or *FAE1* (Supplementary Fig. 11A). To seek further insight into the lipid biochemical phenotype of the *lec1* endosperm, we investigated the expression profiles of key enzyme genes of the cis-ω-7 fatty acid synthesis pathway. Our results showed that *MYB115* and *MYB118* had lower expression levels in *lec1* endosperm than WT at the cellular endosperm (Fig. 4B and E). However,

*AAD2* and *AAD3* did not exhibit drastic reduction in *lec1* mutant endosperm (Fig. 5G), explaining the similar total cis-ω-7 fatty acid content level between WT and *lec1* (Fig. 5E). These results suggest that LEC1 in the endosperm unlikely exerts significant control over the expression of desaturases and cis-ω-7 fatty acid production. In contrast, the expression of *bZIP67* was significantly reduced and the downstream gene *FAE1* was drastically repressed in the *lec1* endosperm, resulting in a decreased proportion of the very long chain cis-ω-7 C20:1 in TAG (Fig. 5G). In addition, we examined the expression levels of *MYB115*, *MYB118*, *AAD2*, *AAD3*, *bZIP67*, and *FAE1* in *lec1* seeds from 1 DAP to maturation stage (Supplementary Fig. 11B)[18] and the results were in agreement with our transcriptome analysis results. These findings thus establish the role of LEC1 in regulating storage lipid synthesis in endosperm (Fig. 5H).

## Discussion

The double fertilization event is possibly one of the driving factors of flowering plants being one of the predominant life forms on earth[14]. Deciphering the detailed coordination in the development of the two fertilization products is an important topic in seed biology. Endosperm supports embryo

growth by providing nutrients and growth regulators, which is critical to generate viable seed[38]. LEC1 functions as a central regulator in diverse development processes in seeds[17–19]. When *LEC1* is exclusively expressed in the endosperm before CE, LEC1 proteins can mobilize to the embryo to activate processes of seed maturation, thus fully complementing the *LEC1* mutant seed phenotype, such as embryo morphology, seed germination, and storage protein accumulation[20]. What role LEC1 specifically plays in the endosperm, in addition to serving as an endosperm signal molecule in regulating embryo development, remained unclear. A significant challenge to address this question was to obtain sufficient amount of endosperm free from contamination of other tissues. We have adopted a modified INTACT system to gather pure endosperm nuclei for ChIP-Seq analysis to profile the genome-wide occupancy of LEC1 in developing endosperms (Fig. 1B–E, Supplementary Figs. 2, 3).

The combined results from ChIP-Seq and RNA-Seq data analyses showed that LEC1 directly and indirectly regulates genes involved in a set of biological processes in endosperm through activation or repression (Figs. 2, 3, Supplementary Fig.4 and 5), and this mode of actions is in part due to the binding specificities of the transcription factors with which it interacts[18]. In syncytial endosperm, we found that LEC1 directly activated a group of endosperm-expressed *AGL* genes (Fig. 2C-D), supporting our hypothesis that LEC1 is associated with different in vivo patterns of DNA binding in the endosperm and the embryo. *AGL62* was identified as one of the LEC1-regulated *AGLs* and its expression is required to promote auxin synthesis and repress endosperm CE[8,9]. The down-regulation of endosperm auxin biosynthesis genes, i.e., *YUC10*, *YUC11*, *TAA1*, and *TAR1*, and the expressional elevation of a group of genes involved in auxin transport and signaling in the *LEC1* endosperm (Fig. 2H, Supplementary Fig. 5B) suggest that *LEC1* is required to modulate auxin biosynthesis, transport, and signaling. Signaling and homeostasis of auxin involve regulation of both its de novo synthesis and transport. The changes in gene expression of auxin transport in *LEC1* endosperm could simply be a feedback response to the compromised auxin biosynthesis, under such scenario, the role of LEC1 would then be indirect, rather than direct. Future work are needed to understand how all these fit in together during seed development. We also found evidence that *LEC1* was involved in restricting nutrient flow from the endosperm to the embryo by regulating sugar transporter genes (Fig. 2G), as well as to repress cell wall formation in early endosperm development (Fig. 2H). After CE, a major shift of LEC1-driven regulatory networks occurs in the endosperm. Our results uncovered distinct sets of LEC1-occupied genes in different stages of endosperm development (Figs. 3 and 4A). LEC1-regulated genes after CE were mostly enriched for biological processes related to seed maturation, i.e., plant organ development and lipid storage (Fig. 3). Moreover, we also discover that only a small portion of LTGs identified in embryo is also found in endosperm and the greater majority of LTGs in embryo were involved in a set of biological processes that are different from that in endosperm (Fig. 4A-B). This conclusion is also supported by the distinct sets of DNA motifs discovered in endosperm and embryo. In this study, LEC1-binding DNA motifs identified in the endosperm, i.e., the MDAS, REM19/VRN1, ABI3 DNA motifs, except BPC1, differ from those found in the embryo, i.e., the G-box and the CCAAT-binding motifs (Fig. 4C). LEC1 is an atypical NF-YB subunit of the NF-Y transcription factor that binds the CCAAT DNA motif; however, the CCAAT motif is only significantly enriched in the LTGs of Arabidopsis seeds at bent cotyledon stage (around 9 DAP)[18]. That might explains that CCAAT is not significantly enriched among the LTGs identified in this study. These observations imply that the roles of LEC1 in regulating gene expression in endosperm and embryo are associated with different in vivo patterns of DNA binding in the two sibling tissues. It's speculated that the same transcription factor shows different DNA binding specificities when it interacts with tissue-specific protein complexes. A similar regulatory mechanism has also been shown in the dual role of a MADS-domain protein FRUITFULL in regulating floral transition and pistil development[39].

After CE, endosperm quickly undergoes DE to facilitate embryo expansion[13]. Embryo invasion triggers a physical stress, serving as a signal, in the surrounding endosperm to initiate programmed cell death[2]. We observed a persistent endosperm with an arrested embryo in *LEC1* mutant seed at even 10 DAP (Fig. 5A), implying a coordinative role of LEC1 in the endosperm-embryo communication. Unlike dicots such as Arabidopsis, most monocots retain endosperm in mature seeds. Understanding LEC1's function in the endosperm of different seed types remains an interesting question. In rice, for example, two *LEC1* homologs, *OsNF-YB9* and *OSNF-YB7*, have been identified[23]. *OsNF-YB9* is specifically expressed in the endosperm, whereas *OSNF-YB7* is exclusively expressed in the embryo. Notably, *OsNF-YB9* cannot substitute for *OSNF-YB7* in rice seed development but can fully rescue *lec1* seed defects in Arabidopsis. Further research is needed to explore the distinct roles of these two rice *LEC1* genes in seed development.

In mature seeds, lack of *LEC1* altered the endosperm TAG fatty acid compositions, thereby unveiling a hitherto unknown metabolic phenotypes entirely consistent with the molecular data analyses of LEC1-regulated transcriptome hubs (Fig. 5B–G, Supplementary Figs. 7–10). An endosperm regulatory network of oil and fatty acid composition control focusing on the production of ω-7 fatty acid has been previously proposed[35]. The endosperm specific cis-ω-7 acyl-ACP desaturase genes *AAD2* and *AAD3* are transcriptionally activated by MYB115 and MYB 118, which themselves are subjected to the transcriptional regulation of LEC1. Given that *LEC1* is also expressed in embryo, the conspicuous lack of expression of *AAD2* and *AAD3* in the embryo is advocated to be under epigenetic control. In this study, we found the transcript levels of *AAD2* and *AAD3* being only modestly affected in *lec1* mutant, and accordingly, the level of total cis-ω-7 fatty acids were not reduced (Fig. 5E–G). Hence, the production of total cis-ω-7 fatty acids in *lec1* remained relatively unperturbed, further accentuating the complexity of the LEC1 participated multicomponent regulatory regime in gene regulation. The elongation of the cis-ω-7 C18:1 fatty acid to VLCFA in the cytosol; however, was clearly compromised (Fig. 5F), primarily due to a drastically decreased expression of the *FAE1* that is responsible for the synthesis of cis-ω-7 C20:1 from cis-ω-7 C18:1 in the cytosol. Our data also suggest that the reduced expression of *FAE1* can be tracked at least in part to the regulatory role of LEC1 to *BZIP67*. These findings thus extend the model of lipid biosynthesis network in the endosperm with *LEC1* playing a prominent role at the top.

Our findings provide important insights into the requirement of LEC1 for endosperm development. It appears that the combinatorial nature of LEC1 action would confer more functional specificities while also broadening its target diversity at the same time. A direct functional consequence of such a combinatorial action is that LEC1 can be involved in many seed developmental processes, playing more a "quantitative" role, rather than an essential one. Such an arrangement might have been evolved in seed plants to protect the genetic resilience of their progeny, i.e., seeds, under diverse challenging environments. Further identification and characterization of LEC1 partners will be required in the future to better understand seed development. Moreover, many direct targets of LEC1 in the endosperm can also influence embryo development, which provides a genetic clue in understanding the embryo-endosperm interactions. We thus believe that this work provides a much-needed resource for the plant biology community to investigate the roles of *LEC1* in endosperm development and endosperm-embryo communication in seed development.

## Methods
### Plant materials and growth conditions
*Arabidopsis thaliana* ecotype Columbia-0 (Col-0) and Ler was used as WT. Arabidopsis mutant *lec1-1* (SALK_131219) and *lec1-3* (CS5739) were obtained from the Arabidopsis Biological Resources Center (ABRC). The transgenic line *pLEC1::LEC1-GFP lec1-1* (PLL) and *pLEC1::LEC1-GFP lec1-3* (LLL) were generated from our previous work[20]. Transgenic plants of PLL *pPHE1::mCherry-NLS*, PLL *pACT2::TurboID pPHE1::mCherry-WPP-BLRP* (PLL-PIM), PLL *pACT2:TurboID pZOU:mCherry-WPP-BLRP* (PLL-ZIM) were generated in this work. Seeds were germinated either in soil or on

half-strength MS medium and grown in a growth room (16 h light/8 h dark; 22 °C; 65% humidity).

## Cloning and generation of transgenic plants

To generate the pPHE1::mCherry-NLS construct, mCherry-NLS (with a stop codon after NLS sequence) was cloned with primers Pac1-AvrII-mCherry-F (forward) and Asc1-NLS-mCherry-R (reverse), transferred to the pMDC162 construct. To generate the pACT2::TurboID mCHERRY-WPP-BLRP construct, DNA synthesis of pACT2::TurboID-Nos Terminator with Pme1 restriction size at the 5′ and Asc1 restriction site at the 3′ (Supplementary Table 3) was delivered by GenScript (Piscataway, NJ). The synthesized DNA sequence was then ligated into pMDC123 vector to get the pACT2::TurboID construct. To generate pACT2::TurboID WPP-BLRP, WPP-BLRP-NOS TERMINATOR was amplified with primers Pac1-AvrII-RAN-F (forward) and Spe1-NOS-R (reverse), transferred to the pACT2::TurboID construct. To build the pACT2::TurboID mCHERRY-WPP-BLRP construct, mCHERRY was cloned with primers Pac1-mCherry-F (forward) and AvrII-mCherry-R (reverse), then inserted into the pACT2::TurboID WPP-BLRP. To build the transformation constructs pACT2::TurboID pPHE1::mCHERRY-WPP-BLRP and pACT2::TurboID pZOU::mCHERRY-WPP-BLRP, the promoter regions of PHE1 and ZOU were amplified from Arabidopsis Col-0 genomic DNA, then transferred into pACT2::TurboID WPP-BLRP through BP and LR reactions by following gateway cloning protocol (Thermo Fisher Scientific, CA). Plant transformation experiments were conducted by following floral dipping method. Primer information is listed in Supplementary Table 3.

## Microscopy and imaging

Confocal images were taken using a Zeiss LSM 880 laser scanning microscope with the following setting of excitation/emission wavelengths: DAPI (405/420-480 nm), GFP (488/505-530 nm), and mCherry (543/550-650 nm). Confocal images were analyzed using the imaging software: LSM Image Browser and Fiji ImageJ software.

## Western blot

To detect the presence of TurboID, 0.1 g of siliques was ground into fine powder in liquid nitrogen and suspended in a two-fold volume of lysis buffer. To detect mCherry, 0.2 g of siliques was ground into fine powder in liquid nitrogen and incubated in Honda buffer for nuclei extraction, followed by protein extraction with lysis buffer. Total protein was cleared by removing debris and mixed with SDS loading buffer. Samples were boiled for 5 min before loading into SDS-PAGE gels. Proteins were detected by antibodies raised against TurboID (1:5000,v:v) (Produciton no. AS204440, Agrisera, Sweden) and mCherry (1:1000,v:v) (CAT#ab167453, Abcam, Cambridge, UK). Rabbit IgG was used as secondary antibody (1 µg/ml) (CAT#ab171870, Abcam, Cambridge, UK).Total loading proteins served as loading control.

## Endosperm nuclei purification and ChIP-Seq assays

Plant siliques (~250 mg for each sample) were collected from PLL-PIM (1–3 DAP), PLL-ZIM (5 and 7 DAPs). Endosperm nuclei purification and ChIP-Seq assays were conducted by following an established protocol with slight modifications[25]. Briefly, siliques were ground into fine powder in liquid nitrogen and resuspended in Honda buffer for 15 min at 4 °C. The mixtures were filtered through miracloth (Millipore, MA) twice and lastly through a CellTrics strainer (30-µm) (Sysmex, Germany). Total nuclei were collected after centrifugation at 1500 g for 6 min at 4 °C, followed by re-suspension in PBSB buffer. The suspension was incubated rotating with pre-blocked Dynabeads M-280 streptavidin beads (Invitrogen, Waltham, MA) for 2 h. After incubation, the beads were collected using a magnet rack, then gently resuspended in PBSBt buffer for 15 min at 4 °C. To purify the biotin-labeled nuclei, the beads incubated with PBSBt buffer passed through the column-like separation system at a speed of ~2 ml/min. The purified nuclei with beads were resuspended in PBSB buffer and used directly for ChIP assays.

Purified nuclei were fixed in 1% formaldehyde and incubated on ice for 8 min, followed by adding glycine for 5 min incubation. Beads were collected and resuspended in the nuclei lysis buffer, then the suspension was subjected to sonication. The sonicated chromatin solution was diluted with ChIP dilution buffer and was further split into the necessary number of portions. The aliquot was incubated with anti-GFP (1:100,v:v) (CAT#ab290, Abcam, Cambridge, UK) overnight at 4 °C with gentle rotation, followed by incubation with Dynabeads protein A (Invitrogen, Waltham, MA) for 2 h. The beads were collected with a magnetic rack and washed by low salt wash buffer, high salt wash buffer and TE buffer. ChIP-DNA was purified from the washed beads by using a Ipure Kit v2 (Diagenode, NJ). ChIP-DNA was quantified by using a Quant-IT dsDNA HS Assay Kit on the Qubit system. Two biological replicates of ChIP-DNA samples were pooled together for one library preparation. The sequencing library was prepared using Accel-NGS 2S Plus DNA Library Kit (Swift, UK) and subsequently used for Illumina NOVAseq 6000 pair-end sequencing. Two independent replicates were conducted for each ChIP-seq, and three biological replicates were performed for ChIP-qPCR. The primers used for ChIP-qPCR are listed in Supplementary Table 3.

## ChIP-Seq data analyses

Raw reads were uploaded to Galaxy (http://usegalaxy.org/). Sequences were trimmed with Trim Galore (Galaxy v 0.6.7)[40] before mapping to the Arabidopsis reference genome (TAIR10) with Bowtie2[41] for Illumina (Galaxy v 2.5.0) for DNA sequencing with pair end and other settings as default. Unmapped and PCR duplicates were filtered out using SAMtools[42], only uniquely and perfectly mapped reads were retained for further analysis. The bigwig format files were generated by bamCoverage with "bin size 10" and "normalize to RPKM (reads per kilobase per million)" in Deeptools[43] for visualization using Integrative Genomics Viewer[44]. Peak calling was conducted using MACS2 (Galaxy v 2.2.7.1) with default parameters[45]. Sequencing reads from Input-DNA were used as controls. Common peaks shared by two biological replicates were determined by IDR (Galaxy v 2.0.3) with values less than 0.05 for further analysis[46]. The ChIPseeker(Galaxy v 1.18.0) was used to assign peaks to proximal genes[47]. Deeptools: ComputeMatrix (Galaxy v 3.5.1) and plotProfile (Galaxy v 3.5.1) were used to compare the mean occupancy density of signals at defined loci. To identify DNA sequence motifs enriched for the LEC1-occupancy sites, 300 bp sequence surrounding each peak summit (150 bp upstream and downstream, respectively) was extracted and searched for enriched DNA motifs using the CentriMo motif analysis with the input motifs database: JASPAR CORE (2022) plants[48].

## GO analysis

GO analyses for enrich biological processes were conducted with online tool (http://geneontology.org/). GO terms were identified with indication of "overrepresentation" by fold enrichment and p-values.

## RNA isolation, RNA-Seq analysis, and RT-qPCR

Total RNA was extracted from hand-dissected WT and lec1-1 endosperms at NP (3 DAP), CE (5 DAP), and DE (7 DAP), separately, using a RNeasy plant mini kit (Qiagen, Germany) according to the manufacturer's instruction. Twenty seeds were used for each RNA extraction and three biological replicates were prepared for RNA-Seq with Illumina NOVAseq 6000 pair-end sequencing. Raw reads were trimmed with Trim Glore before mapping to the TAIR 10 Arabidopsis genome using STAR with default settings[49]. Mapped reads were used to generate feature counts for differential gene expression analysis with DESeq2[50]. Genes with at least 1.5-fold change in expression (adjusted p < 0.05) were considered as differentially expressed.

RNA for each sample was used in reverse transcription reactions using a Reverse Transcription Kit (Qiagen, Canada). For each quantification-PCR, SYBR Green master mixes kit (Thermo Fisher, Canada) with LEC1-specific and CACS (endogenous control) primers (Supplementary Table 3) were used to conduct qPCR reactions on an Applied Biosystems StepOne Real-Time PCR System (Thermo Fisher, Canada). Public Arabidopsis WT

and *lec1-1* seed transcriptome analysis was downloaded from published supplementary files[18]. RMA-normalized and averaged signals were used to calculate relative expression level of lipid-related genes in *lec1-1* compared to WT. The lists of seed-specific and seed coat-specific genes retrieved from Belmonte et al. were used for tissue contamination analysis[22].

## Semi-thin sectioning and staining

Developing seeds from different genotypes were fixed in ice-cold PEM buffer (50 mm PIPES, pH 6.9) with 4% w/v of paraformaldehyde for 1 h under vacuum. The samples were then dehydrated in a gradient of ethanol (10% to 100%, increased by 10%) and subsequently included in a gradient of Spurr's Low Viscosity plastic (Sigma, Canada) mixed with ethanol (plastic to ethanol, 1:3, 1:1, 3:1) until reaching 100% plastic. The Spurr plastic were prepared by mixing the components of ERL 5 g, DER 3 g, NSA 13 g, dimethylaminoethanol 0.4 g. Samples infiltrated with 100% plastic were placed in embedding molds and polymerized in an oven (60 °C) overnight. 0.5 μm sections were then cut with an ultra-microtome (Ultracut, Reichert-Jung) and mounted on glass slides. Sections were dried out on a hot plate (60 °C) before staining with 1% w/v Toluidine blue O for 5 min. Images of the sections were taken with a Zeiss AxioPlan.

## Seed lipid analysis

The endosperm with seed coat fractions and the embryos collected from 15 seeds of each genotype were heated in 1 ml of 5% $H_2SO_4$ (v/v) in methanol at 80 °C for 2 h. The fatty acid methyl esters were extracted with hexane and determined by GC with a flame ionization detector on a DB-23 column. Fatty acid C15:0 was used as internal standard to calculate the total oil content of the embryo and the endosperm. The endosperm was analyzed with the seed coat attached[34,35].

## Statistics and reproducibility

All statistical analyses were performed using R v.4.1.2 (https://www.r-project.org). Comparisons between two groups were conducted using the Student's *t*-tests, with a significance threshold of $p < 0.05$. One-way ANNOVA analyses and post-hoc Tukey tests were conducted to determine the significant difference of multiple groups. All analysis presented in this study were based on a minimum sample size of three replications. Each experiment was repeated at least three times to ensure reproducibility.

## Reporting summary

Further information on research design is available in the Nature Portfolio Reporting Summary linked to this article.

## Data availability

Data used in this study is available as described in the methods section and include in the article and its supplementary materials. The uncropped blot images for figures are available in the Supplementary Fig. 2B. All ChIP-Seq and RNA-Seq raw datasets and process data files generated from this study were deposited into the National Center for Biotechnology Information under project ID PRJNA1100724. The numeric source data for graphs are available in Supplementary Data 5. All other relevant data are available from the corresponding authors upon reasonable request.

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

## Acknowledgements
We thank ABRC for providing the mutant seeds used in this study. We also thank Dr. Daiqing Huang for critical reading of the manuscript and Dr. Teagen Quilichini for technical advise. This work was supported by the Base Project (A1-021754-03 to J.S. and J.Z.) as its contribution to the Aquatic and Crop Resource and Development Center of the National Research Council; and grants from Agriculture and Agri-Food Canada and National Science and Engineering Research Council of Canada (RGPIN/04625-2017; to Y.C.). X.X. was supported by a graduate fellowship from the Chinese Scholarship Council. The authors declare no conflict of interest.

## Author contributions
J.Z., Y.C., and J.S. conceived and coordinated the study; J.S., Y.C., and J.Z. designed the experiments; J.S. and X.X. designed and built the INTACT constructs. J.S. and I.M. conducted plant transformation and transgenic line screening; I.M. performed Western blots; J.S. performed endosperm nuclei extraction and ChIP assays, J.S., B.Y., I.M., W.S., and H.Y. conducted data analysis; Y.W. and J.S. conducted microscopy and imaging. J.S. conducted semi-thin sectioning and staining. J.S. and D.X. conducted seed dissection for RNA-Seq; J.S., Y.C., and J.Z. wrote the manuscript.

## Competing interests
The authors declare no competing interests
