## [Transparent peer review file · Communications Biology]

Tracking the genome-wide occupancy of Arabidopsis LEAFY COTYLEDON1 in endosperm development

Corresponding Author: Professor Jitao Zou

Version 0:

Reviewer comments:

Reviewer #1

(Remarks to the Author)

This manuscript entitled "Tracking the genome-wide occupancy of Arabidopsis LEAFY COTYLEDON1 in endosperm development" offers valuable insights into the role of LEC1 in dicot endosperm development and endosperm-embryo interactions.

I think that the framework and experimental design of this manuscript are well-structured, and the results have stimulated extensive reflection on the functions of LEC1. Here, my only comment is that the authors could further enhance the manuscript by discussing the roles of LEC1 in monocots and dicots, particularly in their endosperms, within the Results or Discussion section. For instance, since monocots like rice and maize have unique permanent endosperms, investigating the expression patterns of LEC1 in endosperms of these two crops and comparing it to those in Arabidopsis would provide valuable insights. While this discussion might extend beyond the manuscript's primary focus, given the significant implications of LEC1 for endosperm and embryo development, a moderate exploration of this topic would undoubtedly further enhance the value of this manuscript.

Reviewer #2

(Remarks to the Author)

The manuscript submitted by Song et al., presents novel insights into the role of LEAFY COTYLEDON1 (LEC1) in the control of endosperm development in Arabidopsis. The authors performed a series of tissue-specific chromatin immunoprecipitation (ChIP) and RNA-seq to identify potential target genes of LEC1 in the Arabidopsis endosperm at different developmental stages. Moreover, the authors showed that LEC1 is involved in the control of lipid accumulation in the endosperm, similarly to what is known in the embryo. Due to technical difficulties in isolating endosperm cells, the authors resorted to using a modified version of INTACT, which has been shown to be a reliable method for the isolation of specific cell types in plant systems. I personally found the manuscript well written, and the flow of the experiments and figures followed a cohesive and compelling story. It was easy to follow the goal of each experiment and the leading conclusions.

However, I would like to raise a few points about the analysis presented in this paper. Given the technical challenges of enriching for endosperm cells both using INTACT as well as through manual dissection, the authors would benefit from a more thorough examination of the produced data. This would be important to evaluate the quality of the data produced, which are important to validate many of the conclusions raised by the authors. As this work can be of a great interest for many seed scientists, I listed a few suggestive points below to help the authors to make a better assessment on the identification of LEC1 targets in the seed endosperm. I believe that addressing these points is important to recommend this paper for publication.

- Evaluation for embryo/seed-coat contamination of hand-dissected endosperm for RNA-seq analysis.

As mentioned by the authors, the manual dissection of endosperm cells is technically difficult with high risk of tissue contamination (Line 95,96). This challenge was addressed for the ChIP-Seq analysis using INTACT. However, for the RNA-seq, the authors decided on isolating endosperm manually. Given the high RNA content in the cytoplasm, this decision is understandable; however, thorough validation of the data quality remains essential, especially since many conclusions rely on this data. I would suggest the authors to compare the generated RNA-seq with the laser-capture microdissection (LCM) seed tissue-specific transcriptome data (Belmonte et al. 2013, PNAS (<https://www.pnas.org/doi/10.1073/pnas.1222061110>)).

Specifically, to show that embryo and/or seed coat specific transcripts are not enriched in their transcriptome datasets.

- Validity of LEC1 Bound regions at CE and DE stages

The authors performed INTACT ChIP-Seq to identify bound regions of LEC1 at CE and DE stages. However, the overlap between LEC1 bound regions and DEGs in *lec1* mutants showed to be surprisingly low (less than 1% of DEGs) (Figure 3C&D). The overlap was greater when compared to the LEC1 bound regions at the NP stage (~30% of DEGs). Additionally, the great majority of LEC1 common bound regions at CE and DE stage were mapped at TE regions. The authors stated that pPHE1 promoter used for the INTACT lines is repressed at the cellularization stages of endosperm development (CE and DE) (Line 231-232). This, in addition to the poor overlap between DEGs and LEC1 bound regions, raises the concern on the validity of the identified LEC1 bound sites at CE and DE stages. The authors made the effort to validate the LEC1 bound region at NP stage through qPCR (Figure S4) but not at CE and DE stages. I'm fairly convinced about the data quality of the ChIP-Seq at the NP stage but not the CE and DE stages. I experienced many times that TF ChIP-Seq quality varies a lot in different developmental stages. If the authors cannot provide additional compelling evidence to confirm the validity of LEC1 bound regions at CE and DE stages (besides the ones already presented in this work), I would strongly recommend considering the conversion of Figure 3 to a supplementary figure and adding comments on the potential limitations regarding data validity. This adjustment would be important so that other researchers would use this data with caution in the future. Much like the analysis done in Figure 4, the LEC1 bound sites at NP stage could be used as a proxy for LEC1 bound regions in the endosperm to potentially identified the direct targets of LEC1 and CE and DE stages.

- Comparison between LEC1 targets in the embryo and endosperm LEC1

The authors identified direct target genes of LEC1 in the endosperm and identified that LEC1 target distinct genes at different developmental stages of endosperm development. As the authors mentioned, this is similar to what was previously found in the embryo. The authors provided compelling evidence to show that LEC1 in the endosperm, much like in the embryo, regulate the accumulation of seed lipids through other TFs (bZIP67 and MYB TFs). As the biological processes occurring in endosperm are quite distinct from the ones in the embryo, I believe that comparing the list of bound regions, and direct targets of LEC1 in the endosperm and in the embryo will greatly enrich the value of the work. The list of LEC1 bound regions and target genes in the Arabidopsis embryo have been previously identified (Pelletier et al. 2017, <https://doi.org/10.1073/pnas.1707957114/>). I believe it would be of great interest to compare the list of identified LEC1 target genes in the endosperm to the ones identified in the embryo. Are they the same? How many? What is the biological function of genes uniquely targeted by LEC1 in the endosperm or the embryo? This comparison will further increase the validity of the data generated in this work as well as providing insights into the distinct role of LEC1 in the endosperm that are different/similar to the ones identified in the embryo.

- DNA motif analysis on target genes

The authors provided a series of enriched DNA elements found in LEC1 bound regions (Figure 1). Interestingly, the DNA elements found in the bound regions at NP stage are not the putative DNA elements bound by LEC1 or LEC1 known partners. Because the analysis was done using all the regions bound by LEC1 (> 8000 at NP stage), this could lead to challenges in identifying meaningful DNA elements important to specify LEC1 function in the endosperm. The motif identified in particular (TATA) is relatively like a general TATA box that are enriched in regions near transcription start sites. In this sense, it is rather difficult to assume if these elements are important to specify LEC1 occupancy or are just enriched due a positional bias of the provided sequences. I would suggest the authors to perform a similar analysis in the genes that are noted as direct targets of LEC1 at NP stage (bound and DEG). The reduction on the number of genes can potentially help to identify more meaningful DNA elements that are specifying LEC1 occupancy. Additionally, it would be important to provide background sequences to the Centrimo test, to potentially exclude DNA elements that are generally enriched in regions near the TSS, like the TATA Box.

I would appreciate if those points could be addressed as it would significantly improve the assessment of the data generated in this work. Something that I believe would be of a great interest for many seed scientists.

Additionally, I had a few other minor suggestions to improve the manuscript (listed below).

- Please provide the bed files for the peak coordinates for the LEC1 bound regions identified at NP, CE and DE stages.
- For the DEG lists, please include a full description on the fold changes as well as the FDR (or adjusted pvalue) for each DEG identified.
- The profile plots in figure S3 show an almost identical profile between CE and DE datasets. Yet the bound genes between CE and DE are fairly distinct (~40% difference). Please make sure the profile plots are not duplicated.
- Some of number of genes in the Venn Diagrams in S5 are not adding up, please double-check the Venn diagrams.
- On Figure 1C, is the profile plot showing the average signal around the peak summit? What is the purpose of this plot? For every peak, you would expect to have a stronger read signal around the peak summit. This is not remarkable. Perhaps this profile would have more meaning if the signal was plotted against the LEC1 bound genes transcription start site? Additionally, what is the meaning of the grey line in these plots? Are they coming from the input (please clarify in the legends)? I would suggest to either remove the plot or replace it with the signal distribution along the transcription start site.

Reviewer #3

(Remarks to the Author)

In this study, the authors used a modified INTACT system to purify endosperm nuclei in Arabidopsis and performed ChIP-seq at three stages. The results showed that LEC1 controls various biological processes in the endosperm. Additionally, the mutation of LEC1 affects endosperm fatty acid composition. However, I have a few concerns:

1. The quality of the ChIP results needs further evaluation. Are there any enriched motifs similar to the known motifs of the NFY family transcription factors? Can the LEC1 protein itself be detected by western blot in the immunoprecipitation (IP) process of the ChIP experiment?
2. Many of the ChIP peaks shown in the IGV plots are very noisy in Figures 2 and S4. For example, no obvious peak for bZIP62 can be seen in Figure S4. The authors need to present the independent replicates for further evaluation.
3. Are there any shared genes bound by LEC1 in both the endosperm and the embryo?
4. Regarding the endosperm defects shown in Figure 5A and 5C-D, the authors should check these phenotypes in both the PLL complementary line and the independent *lec1* mutant line *lec1-4* to exclude the effects of cryptic mutations.

Version 1:

Reviewer comments:

Reviewer #1

(Remarks to the Author)

Well done. This is an insightful article that offers valuable new perspectives related to the well-known gene LEC1 in Arabidopsis.

Reviewer #2

(Remarks to the Author)

I appreciate the time and effort you put into addressing the feedback. The revisions have greatly improved the clarity and overall quality of the paper, and I'm satisfied with the changes. The manuscript is now much stronger, and I have no further concerns.

Congratulations to the authors on their revised manuscript!

Well done!

Reviewer #3

(Remarks to the Author)

All of my concerns have been addressed.

Response to Reviewers' Comments

We sincerely appreciate the the insightful and constructive feedback from the reviewers. Reviewer #1 suggested incorporating additional discussion, which we have carefully addressed. Reviewer #2 expressed confidence in our experimental strategy and results while suggesting further analyses to strengthen the manuscript, which we have undertaken accordingly. Reviewer #3 provided helpful comments and suggestions regarding the data quality check, which we have thoroughly considered. These suggestions have proved to be very helpful in improving the manuscript. A detailed, point-by-point response to each comment is provided below:

Referee expertise:

Referee #1: endosperm, transcription factors

Referee #2: LEC1, seed development, ChIP-seq, RNA-seq

Referee #3: endosperm, transcription factors, ChIP-seq, RNA-seq

Reviewers' comments:

Reviewer #1 (Remarks to the Author):

This manuscript entitled " Tracking the genome-wide occupancy of Arabidopsis LEAFY COTYLEDON1 in endosperm development" offers valuable insights into the role of LEC1 in dicot endosperm development and endosperm-embryo interactions.

I think that the framework and experimental design of this manuscript are well-structured, and the results have stimulated extensive reflection on the functions of LEC1. Here, my only comment is that the authors could further enhance the manuscript by discussing the roles of LEC1 in monocots and dicots, particularly in their endosperms, within the Results or Discussion section. For instance, since monocots like rice and maize have unique permanent endosperms, investigating the expression patterns of LEC1 in endosperms of these two crops and comparing it to those in Arabidopsis would provide valuable insights. While this discussion might extend beyond the manuscript's primary focus, given the significant implications of LEC1 for endosperm and embryo development, a moderate exploration of this topic would undoubtedly further enhance the value of this manuscript.

Response: We appreciate the reviewer's insightful suggestions to expand the discussion on the roles of LEC1 in monocots and dicots. In the revised manuscript, we have incorporated additional discussion (lines 377-385), which now states:

"Unlike dicots such as Arabidopsis, most monocots retain endosperm in mature seeds. Understanding LEC1's function in the endosperm of different seed types remains an interesting question. In rice, for example, two LEC1 homologs, OsNF-YB9 and OSNF-YB7, have been identified. OsNF-YB9 is specifically expressed in the endosperm, whereas OSNF-YB7 is exclusively expressed in the embryo. Notably, OsNF-YB9 cannot substitute for OSNF-YB7 in rice seed development but can fully rescue lec1 seed defects in Arabidopsis. Further research is needed to explore the distinct roles of these two rice LEC1 genes in seed development."

Reviewer #2 (Remarks to the Author):

The manuscript submitted by Song at al, presents novel insights into the role of LEAFY

COTYLEDON1 (LEC1) in the control of endosperm development in Arabidopsis. The authors performed a series of tissue-specific chromatin immunoprecipitation (ChIP) and RNA-seq to identify potential target genes of LEC1 in the Arabidopsis endosperm at different developmental stages. Moreover, the authors showed that LEC1 is involved in the control of lipid accumulation in the endosperm, similarly to what is known in the embryo. Due technical difficulties in isolating endosperm cells, the authors resorted in using a modified version of INTACT, which shown to be a reliable method for the isolation of specific cell types in plant systems. I personally found the manuscript well written, and the flow of the experiments and figures followed a cohesive and a compelling story. It was easy to follow the goal of each experiment and the leading conclusions.

However, I would like to raise a few points about the analysis presented in this paper. Giving the technical challenges of enriching for endosperm cells both using INTACT as well as through manual dissection, the authors would benefit from a more thorough examination of the produced data. This would be important to evaluate the quality of the data produced, which are important to validate many of the conclusions raised by the authors. As this work can be of a great interest for many seed scientists, I listed a few suggestive points below to help the authors to make a better assessment on the identification of LEC1 targets in the seed endosperm. I believe that addressing these points is important to recommend this paper for publication.

- *Evaluation for embryo/seed-coat contamination of hand-dissected endosperm for RNA-seq analysis.*

As mentioned by the authors, the manual dissection of endosperm cells is technically difficult with high risk of tissue contamination (Line 95,96). This challenge was addressed for the ChIP-Seq analysis using INTACT. However, for the RNA-seq, the authors decide on isolating endosperm manually. Given the high RNA content in the cytoplasm, this decision is understandable; however, thorough validation of the data quality remains essential, especially since many conclusions rely on this data. I would suggest the authors to compare the generated RNA-seq with the laser-capture microdissection (LCM) seed tissue-specific transcriptome data (Belmonte et al. 2013, PNAS (<https://www.pnas.org/doi/10.1073/pnas.1222061110>)). Specifically, to show that embryo and/or seed coat specific transcripts are not enriched in their transcriptome datasets.

Response: We thank the reviewer for highlighting the importance of validating data quality and for suggesting a method to assess potential tissue contamination in our RNA-seq dataset. To address this, we retrieved lists of embryo-specific and seed coat-specific genes from Belmonte et al. 2013, PNAS and conducted transcript enrichment analyses. We found that these genes were not significantly enriched in our transcriptome datasets, indicating minimal tissue contamination in the endosperm RNA samples. These findings are presented in the revised Extended Data Fig. 4B-D, with the corresponding description now included in the manuscript:

“Notably, we retrieved embryo-specific and seed coat-specific genes from previous studies²³ and found that their average transcript levels were not significantly enriched in any of the datasets (Extended Data Fig. 4B-D). This suggests minimal tissue contamination in the endosperm RNA samples.”, please see lines 133-136.

- *Validity of LEC1 Bound regions at CE and DE stages*

The authors performed INTACT ChIP-Seq to identify bound regions of LEC1 at CE and DE stages. However, the overlap between LEC1 bound regions and DEGs in lec1 mutants showed to be surprisingly low (less than 1% of DEGs) (Figure 3C&D). The overlap was greater when

compared to the LEC1 bound regions at the NP stage (~30% of DEGs). Additionally, the great majority of LEC1 common bound regions at CE and DE stage were mapped at TE regions. The authors stated that pPHE1 promoter used for the INTACT lines is repressed at the cellularization stages of endosperm development (CE and DE) (Line 231-232). This, in addition to the poor overlap between DEGs and LEC1 bound regions, raises the concern on the validity of the identified LEC1 bound sites at CE and DE stages. The authors made the effort to validate the LEC1 bound region at NP stage through qPCR (Figure S4) but not at CE and DE stages. I'm fairly convinced about the data quality of the ChIP-Seq at the NP stage but not the CE and DE stages. I experienced many times that TF ChIP-Seq quality varies a lot in different developmental stages. If the authors cannot provide additional compelling evidence to confirm the validity of LEC1 bound regions at CE and DE stages (besides the ones already presented in this work), I would strongly recommend considering the conversion of Figure 3 to a supplementary figure and adding comments on the potential limitations regarding data validity. This adjustment would be important so that other researchers would use this data with caution in the future. Much like the analysis done in Figure 4, the LEC1 bound sites at NP stage could be used as a proxy for LEC1 bound regions in the endosperm to potentially identified the direct targets of LEC1 and CE and DE stages.

Response: We thank the reviewer for the helpful suggestion, which has greatly improved our manuscript. We agree with the recommendation to use only the LEC1 bound sites at NP to identify LEC1 targets at CE and DE. In the revised manuscript, we now state:

“Comparatively, 1,721 and 1,724 LOGS were detected at the CE and DE stages, respectively (Extended Data Table 1). The unexpectedly low number of LOGS at CE and DE led us to focus our research on the NP-stage ChIP-seq dataset.” (lines 121-124)

In addition, we have removed the previous Fig.3, which showed the overlap between RNA-seq and ChIP-seq data at CE and DE, respectively. Instead, we have replaced Fig. 3 with new analyses, the updated description and conclusion are now provided in the Result section under “Evidence for a major shift of LEC1-driven regulatory networks in the endosperm after cellularization”. please see lines 192-220.

• *Comparison between LEC1 targets in the embryo and endosperm LEC1*

The authors identified direct target genes of LEC1 in the endosperm and identified that LEC1 target distinct genes at different developmental stages of endosperm development. As the authors mentioned, this is similar to what was previously found in the embryo. The authors provided compelling evidence to show that LEC1 in the endosperm, much like in the embryo, regulate the accumulation of seed lipids through other TFs (bZIP67 and MYB TFs). As the biological processes occurring in endosperm are quite distinct from the ones in the embryo, I believe that comparing the list of bound regions, and direct targets of LEC1 in the endosperm and in the embryo will greatly enrich the value of the work. The list of LEC1 bound regions and target genes in the Arabidopsis embryo have been previously identified (Pelletier et al. 2017, <https://doi.org/10.1073/pnas.1707957114/>). I believe it would be of great interest to compare the list of identified LEC1 target genes in the endosperm to the ones identified in the embryo. Are they the same? How many? What is the biological function of genes uniquely targeted by LEC1 in the endosperm or the embryo? This comparison will further increase the validity of the data generated in this work as well as providing insights into the distinct role of LEC1 in the endosperm that are different/similar to the ones identified in the embryo.

Response: The new analyses have been instrumental in enhancing the quality of the manuscript. As suggested by the reviewer, we retrieved the list of 554 LEC1 target genes (LTGs) in the embryo from Pelletier et al. (2017) to compare with the LEC1 target genes in endosperm generated in this study. Our analyses revealed that 74 out of 554 LTGs identified in the embryo were also present in the endosperm, as shown in the revised Fig. 4A.

To further investigate the biological significance of these genes, we performed GO analyses to examine the functional categories of embryo-unique and endosperm-unique LTGs, as presented in Fig. 4B. The results indicated that:

“As previously reported, LTGs in embryo were mostly enriched for GO terms related to cell wall modification, photosynthesis, and respond to abiotic stimulus¹⁸. In the endosperm, NP-specific LTGs were primarily enriched for GO terms related to secondary cell wall biogenesis, glucosinolate metabolic, regulation of growth, and respond to lipid; however, CE- and DE-specific LTGs were enriched for GO terms related to ABA metabolism and tissue development. The 74 LTGs shared between the embryo and endosperm were involved in fatty acid biosynthesis, somatic embryogenesis, and seed dormancy.”

These findings are now incorporated into the revised manuscript (lines 234-242).

- *DNA motif analysis on target genes*

The authors provided a series of enriched DNA elements found in LEC1 bound regions (Figure 1). Interestingly, the DNA elements found in the bound regions at NP stage are not the putative DNA elements bound by LEC1 or LEC1 known partners. Because the analysis was done using all the regions bound by LEC1 (> 8000 at NP stage), this could lead to challenges in identifying meaningful DNA elements important to specify LEC1 function in the endosperm. The motif identified in particular (TATA) is relatively like a general TATA box that are enriched in regions near transcription start sites. In this sense, it is rather difficult to assume if these elements are important to specify LEC1 occupancy or are just enriched due a positional bias of the provided sequences. I would suggest the authors to perform a similar analysis in the genes that are noted as direct targets of LEC1 at NP stage (bound and DEG). The reduction on the number of genes can potentially help to identify more meaningful DNA elements that are specifying LEC1 occupancy. Additionally, it would be important to provide background sequences to the Centrimo test, to potentially exclude DNA elements that are generally enriched in regions near the TSS, like the TATA Box.

Response: As suggested by the reviewer, we have re-conducted DNA motif analyses, this time focusing exclusively on the direct targets of LEC1. This analysis led to the identification of a new set of DNA motifs. The revised manuscript now states:

“The MADS, BPC1, REM19/VRN1, and ABI3 cis-regulatory elements were significantly overrepresented in all endosperm LTGs across the different stages. The BPC1 motif was the only one also identified in embryo-LTGs. Notably, the CCTTA-binding sequence and G-box motifs, which were reported in the embryo, were not significantly enriched in endosperm.” (lines 251-255).

Additionally, previous research has shown that the CCAAT-binding sequence motif is only detected in later seed development (~9 DAP). These findings are now incorporated into the revised Fig. 4C.

I would appreciate if those points could be addressed as it would significantly improve the assessment of the data generated in this work. Something that I believe would be of a great interest for many seed scientists.

Additionally, I had a few other minor suggestions to improve the manuscript (listed below).

- *Please provide the bed files for the peak coordinates for the LEC1 bound regions identified at NP, CE and DE stages.*

Response: We have deposited the bed files into the NCBI database (GEO accession number GSE291311).

- *For the DEG lists, please include a full description on the fold changes as well as the FDR (or adjusted pvalue) for each DEG identified.*

Response: The information on fold change and FDR has been added to the list.

- *The profile plots in figure S3 show an almost identical profile between CE and DE datasets. Yet the bound genes between CE and DE are fairly distinct (~40% difference). Please make sure the profile plots are not duplicated.*

Response: We have revised the extended figures.

- *Some of number of genes in the Venn Diagrams in S5 are not adding up, please double-check the Venn diagrams.*

Response: We have doubled checked the numbers in each Venn diagram in the revised version of figures.

- *On Figure 1C, is the profile plot showing the average signal around the peak summit? What is the purpose of this plot? For every peak, you would expect to have a stronger read signal around the peak summit. This is not remarkable. Perhaps this profile would have more meaning if the signal was plotted against the LEC1 bound genes transcription start site? Additionally, what is the meaning of the grey line in these plots? Are they coming from the input (please clarify in the legends)? I would suggest to either remove the plot or replace it with the signal distribution along the transcription start site.*

Response: As per the reviewer's suggestion, we have removed the profile plot displaying the average signal around the peak summit. Instead, we have provided a new profile plot that illustrates the signal distribution along the transcription start site (TSS) (Fig. 1G).

Reviewer #3 (Remarks to the Author):

In this study, the authors used a modified INTACT system to purify endosperm nuclei in Arabidopsis and performed ChIP-seq at three stages. The results showed that LEC1 controls various biological processes in the endosperm. Additionally, the mutation of LEC1 affects

endosperm fatty acid composition. However, I have a few concerns:

1. The quality of the ChIP results needs further evaluation. Are there any enriched motifs similar to the known motifs of the NFY family transcription factors? Can the LEC1 protein itself be detected by western blot in the immunoprecipitation (IP) process of the ChIP experiment?

Response: We greatly appreciate the reviewers' thoughtful concerns and feedback. In the revised manuscript, we re-analyzed the DNA motifs enriched in the LEC1 direct targets, and the identified motifs are shown in the revised Fig. 4C. We compared the list of identified motifs with those previously reported to be enriched in LEC1 targets in the embryo (Pelletier et al., 2017). We have provided the following description in the Results section. The Results now states:

“The MADS, BPC1, REM19/VRN1, and ABI3 cis-regulatory elements were significantly overrepresented in all endosperm LTGs across the different stages. The BPC1 motif was the only one also identified in embryo-LTGs. Notably, the CCTTA-binding sequence and G-box motifs, which were reported in the embryo, were not significantly enriched in endosperm.” (lines 251-255).

Additionally, due to the lack of a commercialized available LEC1 antibody, we were unable to directly examine the LEC1 protein in the IP-process of the ChIP experiment. Instead, we utilized an anti-GFP antibody to capture LEC1-GFP associated chromatin/DNA fragments. This anti-GFP antibody used in this study has been well validated and is known to work efficiently in ChIP assays across multiple studies. To further confirm the specificity of LEC1-GFP binding, we performed independent ChIP-qPCR experiments, using anti-IgG as a control.

2. Many of the ChIP peaks shown in the IGV plots are very noisy in Figures 2 and S4. For example, no obvious peak for bZIP62 can be seen in Figure S4. The authors need to present the independent replicates for further evaluation.

Response: We thank the reviewer for pointing out the noise in the original figures. In response, we have re-analyzed the data and updated the figures (Fig. 1H, Fig. 2D & 2F). Additionally, bZIP62, which was used as a non-LEC1-binding control, is shown in Fig. 1H and Fig. 1I.

To clarify any confusion, we have provided the following description in the revised manuscript:

“bZIP62, which was not identified in the LOGS dataset, was used as a non-LEC1 binding control for validation (Fig. 1H-I).” (lines 120-121).

Furthermore, we have included two independent replicates of ChIP-seq signals in the revised Fig. 1H to enhance data transparency.

3. Are there any shared genes bound by LEC1 in both the endosperm and the embryo?

Response: Yes, the comparison yielded positive results. We compared the LEC1 target genes (LTGs) identified in the endosperm with those previously identified in the embryo. Pelletier et al. (2017) reported 554 LTGs in the embryo, which we retrieved and compared with the LTGs identified in this study. Our analysis revealed that 74 out of 554 LTGs identified in the embryo were also present in the endosperm, as shown in the revised Fig. 4A.

Additionally, we conducted GO analyses to examine the biological processes associated with embryo-specific, endosperm-specific, and shared LTGs. The results indicate that:

“LTGs in embryo were mostly enriched for GO terms related to cell wall modification, photosynthesis, and respond to abiotic stimulus¹⁸. In the endosperm, NP-specific LTGs were primarily enriched for GO terms related to secondary cell wall biogenesis, glucosinolate metabolic, regulation of growth, and respond to lipid; however, CE- and DE-specific LTGs were enriched for GO terms related to ABA metabolism and tissue development. The 74 LTGs shared between the embryo and endosperm were involved in fatty acid biosynthesis, somatic embryogenesis, and seed dormancy.”

These findings are now incorporated into the revised manuscript (lines 235-242).

4. Regarding the endosperm defects shown in Figure 5A and 5C-D, the authors should check these phenotypes in both the PLL complementary line and the independent *lec1* mutant line *lec1-4* to exclude the effects of cryptic mutations.

Response: The *lec1-1* (salk_131219) mutant line used in this study has been previously characterized for its defects in seed development by Tao, et. al (2017; Nature) and in our recent work (song, et al 2021; Nature Communications). The *pLECI:LECI-GFP lec1-1* (PLL) transgenic line is a complementary line that was included in the fatty acid composition analysis. We have provided a semi-thin section of PLL seeds collected at 10 DAP, which demonstrates that the morphological defects in the *lec1* endosperm were restored in PLL, please see lines 265-266 for this information.

Instead of *lec1-4*, we introduced another set of Landsberg seeds. The revised manuscript now states:

*“To confirm that LEC1 is responsible for defects in seed oil content, we analyzed seeds from the *lec1-3*, a T-DNA mutant of LEC1 in the Landsberg (Ler) background, along with wild type Ler and the *pLECI::LECI-GFP lec1-3* (LLL) plants 20. Similar to *lec1-1*, *lec1-3* exhibited elevated endosperm oil content and a significantly reduced ratio of cis- ω -7 C20:1 to cis- ω -7 C18:1, both of which were restored in the complementary line LLL (Extended Data Fig.9).”* (lines 288-293)

These findings are also shown in the revised Extended Data Fig. 9.

Response to Reviewers' Comments

We sincerely appreciate the reviewers' efforts and insightful and constructive feedback again.

Reviewers' comments:

Reviewer #1 (Remarks to the Author):

Well done. This is an insightful article that offers valuable new perspectives related to the well-known gene LEC1 in Arabidopsis.

Reviewer #2 (Remarks to the Author):

I appreciate the time and effort you put into addressing the feedback. The revisions have greatly improved the clarity and overall quality of the paper, and I'm satisfied with the changes. The manuscript is now much stronger, and I have no further concerns. Congratulations to the authors on their revised manuscript! Well done!

Reviewer #3 (Remarks to the Author):

All of my concerns have been addressed.